# Functional hearing and low frequency hearing preservation after cochlear implant surgery is achievable with FLEX electrode arrays: Real world evidence from the MEHS Registry

Uwe Baumann[1], Tobias Weissgerber[1], Andreas Radeloff[2], Ilona Anderson[3], Karin A. Koinig[3], Magdalena Breu[3], Stefano Morettini[3], Vera Lohnherr[4], Joachim Müller[5], Daniel Polterauer-Neuling[5]*

1 ENT/Audiological Acoustics, University Hospital, Goethe University Frankfurt, Frankfurt am Main, Hesse, Germany, 2 Division of Otorhinolaryngology and Research Center Neurosensory Science, Carl von Ossietzky University Oldenburg, Oldenburg, Lower Saxony, Germany, 3 Clinical Research Department, MED-EL Elektromedizinische Geräte GmbH, Innsbruck, Tyrol, Austria, 4 Hals-, Nasen- und Ohrenklinik (Kopfklinik) Im Neuenheimer Feld, University Hospital Heidelberg, Heidelberg, Baden-Württemberg, Germany, 5 Clinic and Polyclinic for Ear, Nose, and Throat Medicine, LMU Klinikum, Munich, Bavaria, Germany

* daniel.polterauer@med.uni-muenchen.de

## Abstract

This study uses data from the MED-EL Hearing Solutions (MEHS) multicenter registry to examine the impact of a series of flexible lateral wall electrode arrays on postoperative low frequency hearing preservation (LFHP) of cochlear implant (CI) users. Participants were members of the MEHS registry who had received a MED-EL CI with a FLEX electrode array. LFHP was evaluated using 2 formulae: the Vienna Consensus (VC), which assesses hearing preservation at 250–1000 Hz; and the Minimum Reporting Standards for Adult Cochlear American Academy of Otolaryngology—Head and Neck Surgery (AAO), which assesses hearing preservation at 125–500 Hz. LFHP was assessed via air conduction preoperatively and between 6 and 36 months postoperatively. LFHP was achievable with all the FLEX arrays evaluated, regardless of length. With the VC, 60% of ears (n = 95) had complete or partial LFHP at 6–12 months postoperatively and 68.4% (n = 19) had complete or partial LFHP at 24–36 months postoperatively. With the AAO, 43.4% of ears (n = 83) had LFHP at 6–12 months postoperatively and 84.2% (n = 19) had functional low frequency residual hearing at 24–36 months postoperatively. In conclusion, implantation with FLEX electrode arrays can preserve functional low-frequency residual hearing up to three years after implantation. The MEHS multicenter registry appears to be a valuable tool for collecting large amounts of real-world data from CI users despite challenges in harvesting usable data.

**Data availability statement:** All relevant data are within the manuscript and its Supporting Information files.

**Funding:** This study was funded by MED-EL Elektromedizinische Geräte G.m.b.H. There is no grant number. The funders assisted in study design, data collection and analysis, and preparation of the manuscript. Decision to publish was a joint decision between all authors.

**Competing interests:** Authors Ilona Anderson, Stefano Morettini, Karin Koinig, and Magdalena Breu are employed at MED-EL Elektromedizinische Geräte G.m.b.H. Uwe Baumann and Daniel Polterauer-Neuling have received travel support & research support from MED-EL GmbH (Austria). Polterauer-Neuling is part of MED-EL's audiological scientific advisory board (MAUWIB). Other authors disclose no conflict of interest. Commercial affiliation does not alter our adherence to PLOS ONE policies on sharing data and materials.

## Introduction

A goal in cochlear implant (CI) surgery is to preserve as much of the CI recipients' preoperative residual hearing as possible because CI users with more residual hearing often enjoy greater benefit than those with little or no residual hearing [1,2] and may allow CI recipients to benefit from future technologies. Preserving residual hearing in the low frequencies is regarded as particularly important, as many CI candidates, especially candidates for electro acoustic stimulation, have some degree of residual hearing at these frequencies. To this end, concepts of "soft" (or atraumatic) CI surgery, first promoted by Lehnhardt [3], have become widely accepted, and surgical techniques and materials that preserve residual hearing and intracochlear structures are being continually developed and improved [e.g., 4]. Indeed, atraumaticity is currently a central focus in CIs, as evidenced by an ever-increasing number of articles on corticosteroid regimes, robotic surgery, surgical techniques, and electrode array designs capable of maximizing hearing preservation [e.g., 5–7]. While many studies have broadened our understanding of residual low-frequency hearing preservation (LFHP), most are limited by their comparatively low number of study participants, as is often the case in studies on CI users. Furthermore, data are difficult to compare between studies due to differences in inclusion/exclusion criteria and in their use of varying definitions of what constitutes successful LFHP.

Recently, some countries have established registries of CI recipients, which have enabled researchers to pool very large amounts of real-world data including participants from a variety of surgeons and hospitals during routine, clinical practice [e.g., 8,9]. Registries are an exciting development in the CI field because they have the potential to enable a greater understanding of issues and, therefore, better evidence-based practices.

The primary aim of this study is to determine LFHP rates in CI recipients who received a FLEX series (MED-EL, Innsbruck, Austria) electrode array. To assess this in a large cohort, independent of surgical technique, we used data from the MED-EL Hearing Solutions (MEHS) Registry. As this study is based on registry data, our objective was descriptive, not causal or explanatory. To provide a fuller view, two classification systems were used: the Vienna Consensus (VC) and the Minimum Reporting Standards for Adult Cochlear American Academy of Otolaryngology—Head and Neck Surgery (AAO) [10].

## Materials and methods

### Ethics committee approval

MEHS Registry data for this investigation were pooled from four clinics in Germany, all of whom received Ethics Committee consent for this study: LMU Klinikum (Ethics number: #17-227), University Hospital Frankfurt (Ethics number: #288/17), Evangelisches Krankenhaus Oldenburg (Ethics number: #2018−110), and Universitätsklinikum Heidelberg (Ethics number: #546/2018). The MEHS Registry procedures were performed in adherence with the standards set in the latest revision of the Declaration

of Helsinki. The MEHS Registry was registered within the clinical trials database (ClinicalTrials.gov: NCT05668338). All CI users provided written informed consent before participating in all procedures.

## Participants

Data extraction from the MEHS registry was accessed and completed for all clinics on 28/06/2022. The MEHS is organized as a prospective anonymized data collection from clinical routine datasets. Clinicians in their respective registry center and as part of their clinical routine had access to information that could identify individual participants during study. Only anonymized data were exported and analyzed.

The inclusion criterion for MEHS is all CI recipients at the participating clinics who consent to be included. For more information on the MEHS, see Baumann et al. [11]. The inclusion criteria for this study differed according to assessment method (VC and AAO). For both methods, individuals were included if they had a FLEX array and unaided pure tone average (PTA) measured via air conduction at ≤12 months preoperatively and at least one assessment 6–36 months postoperatively. For the VC, potential participants were excluded if they did not have preoperative low frequency residual hearing, i.e., had ≥ 111.67 dB HL (average maximum output levels of the audiometers) at 250, 500, and 1000 Hz. For the AAO, potential participants were excluded if their preoperative PTA was ≥ 80 dB HL at 125, 250, and 500 Hz.

## FLEX series electrode arrays

First introduced in 2004, the FLEX series of lateral wall arrays are designed to minimize intra-cochlear trauma during insertion. They are available in various lengths to accommodate different cochlear sizes. Except for the FLEXSOFT, which has a length of 31.5 mm, the numbers after "FLEX" in the title of each array (e.g., the "24" in FLEX24) refer to the length of the array in millimetres. The FLEX24 was formerly named the FLEXEAS. For a more detailed description of FLEX arrays, see Dhanasingh [12].

## Vienna consensus (VC)

The VC was introduced in 2023 by an international group of surgeons who met at a surgical advisory board in Vienna on 10-Jan-2023. The VC focuses on the low-frequency part of the audiogram representing relative LFHP by calculating the average from the measurements obtained at 250, 500, and 1000 Hz; measurements exceeding the maximum audiometer test thresholds to the values given in Table IV in Skarzynski et al. [13] are truncated, to obtain a uniform maximum per frequency. To be assessed via VC, CI recipients must have some preoperative hearing at those frequencies <111.67 dB HL (average maximum output levels of the audiometers). Postoperative scores are classified into Complete LFHP (≤15 dB change), Partial LFHP (>15 to ≤30 dB change), and Complete low frequency hearing loss (LFHL, > 30 dB change). The physiological fluctuation influenced by the status of the CI recipient at testing (e.g., time of day at which the test was performed, hydration status, etc.) is ± 15 dB [14,15]. Therefore, a negative threshold difference of more than 15 dB was defined as some degree of postoperative hearing loss.

## Minimum reporting standards for Adult Cochlear American Academy of Otolaryngology—Head and Neck Surgery (AAO)

A protocol for reporting postoperative LFHP was proposed by Adunka et al. [10] and endorsed by the Implantable Hearing Devices Committee and the Hearing Committee of the American Academy of Otolaryngology—Head and Neck Surgery, hence our referring to it here as the AAO. They advocate reporting residual hearing only for CI users with functionally relevant preoperative hearing, which they define as a PTA of <80 dB HL at 125, 250, and 500 Hz. Postoperatively, air conduction thresholds should be reported individually for each frequency from 125–8000 Hz. If a CI user's postoperative residual

acoustic hearing (as defined by their PTA at 125, 250, and 500 Hz) has become poorer than 80 dB HL, it is considered no longer functionally relevant and, therefore, no longer necessary to report.

### VC and AAO convergence

It is important to remember that the VC and AAO use different frequencies to assess different things and for different purposes. The VC categorizes relative hearing preservation (based on postoperative threshold shifts at 250–1000 Hz). Its purpose is to enable standardized, comparable reporting across centers and studies, regardless of whether the absolute thresholds remain within a functionally usable range. In contrast, the AAO defines if postoperative residual hearing is functional benefit (defined as <80dB HL at 125–500 Hz), especially for EAS use. As such, it a clinically oriented framework. Thus, as the VC and AAO were designed to answer different questions and use different frequencies, agreement or discordance between them is neither expected nor clinically meaningful. Nonetheless, should readers be interested in how many cases the outcomes coincided or diverged between metrics, we compared results on a per participant basis. Results were classified as either a match or a mismatch. A match meant a participant had both complete or partial LFHP on the VC and functional hearing on the AAO or had complete LFHL on the VC and no functional hearing with the AAO. Two types of mismatch were calculated: firstly, compete LFHL on the VC and functional hearing on the AAO and secondly, complete or partial LFHP on the VC and no functional hearing on the AAO.

### Safety and device deficiencies

Safety was assessed by collecting clinical incidents. Within the context of the registry, incidents were monitored on-site and reported according to the incident reporting procedure for approved products. Device deficiencies related to the CI system were eligible for the report during the timeframe considered in the data analysis.

### Statistics

Descriptive statistics were used to report participants' demographic and baseline characteristics and to present the study results. Quantitative data were presented as mean with standard deviation (SD) and/or median with range (minimum and maximum); qualitative data were presented as absolute and relative frequencies. All statistical analyses were performed with Microsoft 365 Excel Version 2208 (Microsoft Corporation, Redmond, WA, USA).

## Results

### Participants

Analysis of the registry yielded 704 CI recipients (823 ears). Of these, 115 participants (122 ears) fulfilled the criteria for the VC and 86 people (96 ears) for the AAO. Over 80% of participants in each cohort received a FLEX28 or FLEXSOFT array. For the selection criteria for each cohort, see Fig 1 (VC) and Fig 2 (AAO); for demographics for each cohort, see Table 1.

It should be noted that for the AAO, there are 47 FLEX28 users but data for 48 PTAs (see Table 1 and Fig 2). This apparent discrepancy is because 1 participant was assessed before and after reimplantation.

### Hearing preservation/functional hearing preservation

For the mean pre- and post-operative hearing thresholds (125–8k Hz) according to array type, see the (S1 Table). Whilst preoperative assessment could take place up to >12 before implantation, in most cases it was done earlier: 61.9% at 1–10 days pre-op, 15.5% at 11 days to 3 months, 13% at 4–6 months, 7.2% at 7–8 months, and 2.1% at 9–12 months.

**VC.** Across all arrays, complete or partial LFHP was achieved in 60.0% of ears at 6–12 months, in 57.4% of ears at 12–24 months, and in 68.4% of ears at 24–36 months. For detailed results according to array and interval, see Table 2.

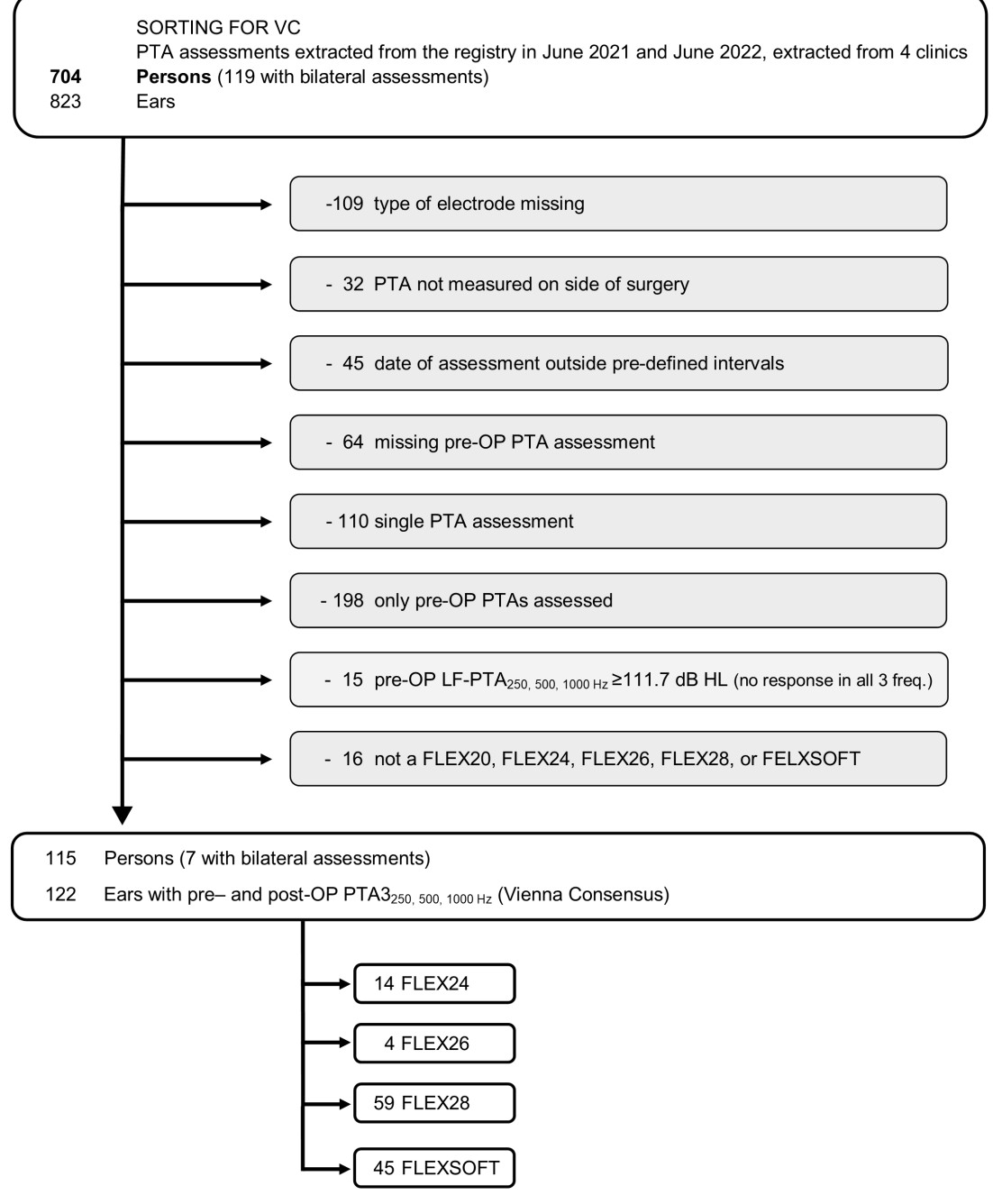

**Fig 1. CONSORT diagram for participants in the VC group.** The number of people (not ears) excluded and reason for exclusion are given in light gray.

**AAO.** Across all arrays, functional hearing was achieved from 43.4% (36/83) of cases at 6 to <12 months to 84.2% (16/19) at 24 to <36 months. For detailed results according to array and interval, see Table 3.

For the mean thresholds from 125 to 8000 Hz at each postoperative interval for each array, see Figs 3–6.

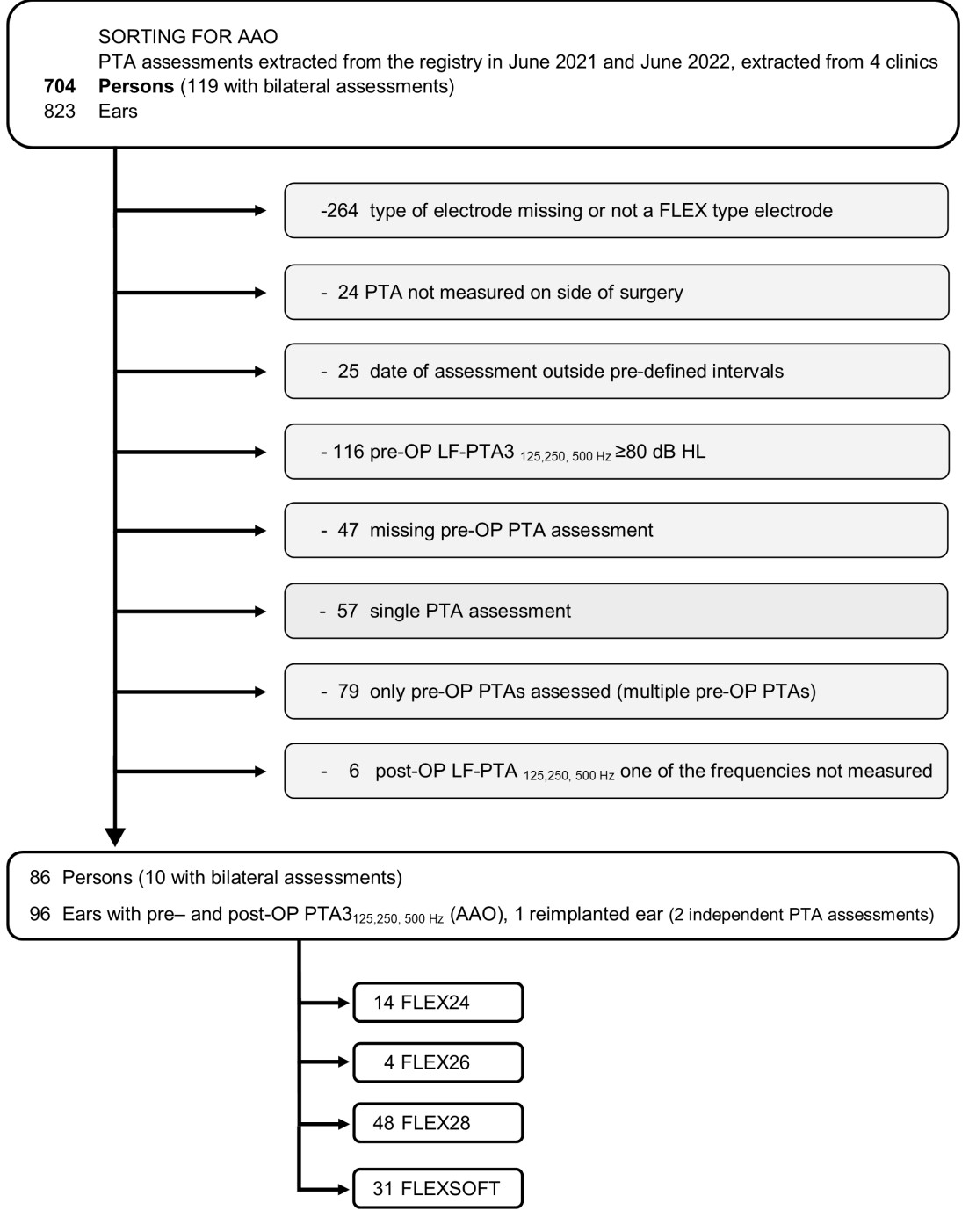

**Fig 2. CONSORT diagram for participants in the AAO group.** The number of people (not ears) excluded and reason for exclusion are given in light gray. It should be noted that there are 47 FLEX28 users but data for 48 PTA assessments. This apparent discrepancy is because 1 participant was assessed before and after reimplantation.

**Table 1. Demographic and baseline characteristics of each study cohort.**

| | VC n (%) | AAO n (%) |
|---|---|---|
| **Participants** | **115** | **86** |
| Ears with CI | **122** | **96** |
| Unilateral | 108 (93.9) | 76 (88.4) |
| Bilateral | 7 (6.1) | 10 (11.6) |
| **Age (years)** | | |
| Mean ± | 59.3 ± 15.7 | 59.7 ± 13.6 |
| Range | 6.4–85.1 | 30.6–85.1 |
| <20 | 3 (2.6) | 0 (0) |
| 20 to <40 | 12 (10.4) | 9 (10.5) |
| 40 to <60 | 43 (37.4) | 35 (40.7) |
| 60 to <80 | 49 (42.6) | 37 (43.0) |
| ≥80 | 8 (7.0) | 5 (5.8) |
| **Sex** | | |
| Female | 69 (60.0) | 48 (55.8) |
| Male | 46 (40.0) | 38 (44.2) |
| Other | 0 (0.0) | 0 (0) |
| **Ear implanted** | | |
| Left (all) | 65 (53.3) | 47 (49.0) |
| Right (all) | 57 (46.7) | 49 (51.0) |
| **Electrode array (per ear)** | | |
| FLEX24 | 14 (11.5) | 14 (14.6) |
| FLEX26 | 4 (3.3) | 4 (4.2) |
| FLEX28 | 59 (48.4) | 47 (49.0) |
| FLEXSOFT | 45 (36.9) | 31 (32.3) |

**VC and AAO convergence.** Results matched in 66.4% cases. Of the 33.6% of cases of mismatch, 14.9% were complete LFHL with the VC but functional hearing preservation with the AAO and 18.7% were complete or partial LFHP with VC but no functional hearing preservation with the AAO. There was no consistent pattern suggesting an overestimation of hearing preservation by either scale. More details can be found in S2 Table.

## Safety and device deficiencies

There was 1 case of reimplantation; however, the cause was not recorded in the Registry. No device deficiencies related to the MED-EL CI system were reported within the timeframe considered in this data analysis. In addition, the data collected concerning the audiological background (PTA preoperatively tested in unaided conditions) shows clinical use in line with the indications as per electrode array variant.

## Discussion

The primary objective of this data analysis was to determine the proportion of ears with LFHP up to 36 months after implantation with a FLEX series array. This objective was met: 704 CI recipients (823 ears) could be extracted and, after the application of inclusion/exclusion criteria, LFHP could be assessed in 115 people (122 ears) via the VC and 86 people (96 ears) via the AAO (see Figs 1 and 2, respectively). This indicates that LFHP is possible with FLEX arrays. These

**Table 2. Hearing preservation results per array and interval using the VC. n are per ear. LFHP = low-frequency hearing preservation; LFHL = low-frequency hearing loss.**

| | n | Complete LFHP ≤15 dB loss | | Partial LFHP >15–30 dB loss | | Complete LFHL >30 dB loss | |
|---|---|---|---|---|---|---|---|
| | | n | % | n | % | n | % |
| **ALL arrays** | | | | | | | |
| 6 to < 12 months | 95 | 31 | 32.6% | 26 | 27.4% | 38 | 40.0% |
| 12 to < 24 months | 54 | 15 | 27.8% | 16 | 29.6% | 23 | 42.6% |
| 24 to < 36 months | 19 | 3 | 15.8% | 10 | 52.6% | 6 | 31.6% |
| **FLEX24** | | | | | | | |
| 6 to < 12 months | 13 | 6 | 46.2% | 6 | 46.2% | 1 | 7.7% |
| 12 to < 24 months | 8 | 3 | 37.5% | 3 | 37.5% | 2 | 25.0% |
| 24 to < 36 months | 7 | 1 | 14.3% | 5 | 71.4% | 1 | 14.3% |
| **FLEX26** | | | | | | | |
| 6 to <12 months | 4 | 1 | 25.0% | 0 | 0.0% | 3 | 75.0% |
| 12 to <24 months | 2 | 1 | 50.0% | 0 | 0.0% | 1 | 50.0% |
| 24 to <36 months | 0 | – | – | 0 | – | 0 | – |
| **FLEX28** | | | | | | | |
| 6 to < 12 months | 43 | 12 | 27.9% | 13 | 30.2% | 18 | 41.9% |
| 12 to < 24 months | 26 | 7 | 26.9% | 9 | 34.6% | 10 | 38.5% |
| 24 to < 36 months | 11 | 1 | 9.1% | 5 | 45.5% | 5 | 45.5% |
| **FLEXSOFT** | | | | | | | |
| 6 to < 12 months | 35 | 12 | 34.3% | 7 | 20.0% | 16 | 45.7% |
| 12 to < 24 months | 18 | 4 | 22.2% | 4 | 22.2% | 10 | 55.6% |
| 24 to < 36 months | 1 | 1 | 100.0% | 0 | – | 0 | – |

relatively large cohorts could be reached thanks to the MEHS Registry, a non-interventional, systematic collection of clinical data in which prospective data are collected from original clinical records.

Results show that (complete or partial) LFHP was achieved with FLEX arrays in 43.4–60% of cases at up to 1 year after first fitting, 52.2–57.4% of cases at up to 2 years after first fitting, and 68.4–84.2% of cases at up to 3 years after first fitting. These results are without controlling for surgical technique, the choice of which is known to affect the likelihood of LFHP [16], or any other factor influencing hearing preservation. Thus, we conclude that LFHP is possible with all of the FLEX arrays.

The data from this descriptive dataset indicates that LFHP is achievable when shorter arrays are used. This is possible because shorter arrays may not reach the low-frequency areas even when fully inserted and so cannot cause insertion trauma or an inflammatory response and fibrosis in these areas. The correlation between deeper insertion depth and poorer LFHP has been supported in multiple studies [e.g., 17–19], while other studies have disputed that deeper insertion depth negatively affects LFHP [20–22]. The present dataset does not enable us to address this question due to multiple factors. Firstly, shorter arrays are often chosen for candidates with more residual hearing, especially for EAS candidates. As is evident in Figs 3–6, the FLEX24 and FLEX26 recipients in the present study (who may have been EAS recipients when this data point was collected) had more preoperative low-frequency residual hearing than FLEX28 and FLEX-SOFT recipients and, therefore, could lose more hearing without suffering complete LFHL (see Figs 3–6). It is very likely, although unproveable, that these individuals received partial, not complete, array insertion. This may have influenced results, especially on the AAO. Secondly, the present observational study did not factor in variables that are known, or hypothesized, to influence LFHP, e.g., surgical technique, cochlear duct length, completeness of array insertion, scala

**Table 3. Hearing preservation results per array and interval using the AAO. n are per ear.**

| Electrode array | N | Functional LFHP < 80 dB HL | | No functional LFHP ≥ 80 dB HL | |
|---|---|---|---|---|---|
| | | n | % | n | % |
| **ALL** | | | | | |
| 6 to < 12 months | 83 | 36 | 43.4% | 47 | 56.6% |
| 12 to < 24 months | 46 | 24 | 52.2% | 22 | 47.8% |
| 24 to < 36 months | 19 | 16 | 84.2% | 3 | 15.8% |
| **FLEX24** | | | | | |
| 6 to < 12 months | 14 | 13 | 92.9% | 1 | 7.1% |
| 12 to < 24 months | 10 | 9 | 90.0% | 1 | 10.0% |
| 24 to < 36 months | 9 | 9 | 100.0% | 0 | – |
| **FLEX26** | | | | | |
| 6 to < 12 months | 4 | 4 | 100.0% | 0 | – |
| 12 to < 24 months | 2 | 2 | 100.0% | 0 | – |
| 24 to < 36 months | 0 | 0 | – | 0 | – |
| **FLEX28** | | | | | |
| 6 to < 12 months | 40 | 16 | 40.0% | 24 | 60.0% |
| 12 to < 24 months | 22 | 12 | 54.5% | 10 | 45.5% |
| 24 to < 36 months | 10 | 7 | 70.0% | 3 | 30.0% |
| **FLEXSOFT** | | | | | |
| 6 to < 12 months | 25 | 3 | 12.0% | 22 | 88.0% |
| 12 to < 24 months | 12 | 1 | 8.3% | 11 | 91.7% |
| 24 to < 36 months | 0 | 0 | – | 0 | – |

tympani volume, age, and sex [21–23]. The use of soft surgical techniques is regarded as especially important, although their use still does not guarantee complete LFHP [5,24]. It could be of substantial future benefit if these data points were recorded in the MEHS. Thirdly, several different surgeons performed the implantations, but it is not known how many. Finally, the sample sizes are small and uneven, especially at later intervals.

Another point of debate is comparing rates of LFHP achieved with perimodiolar arrays and lateral wall arrays. It is commonly thought that lateral wall arrays (e.g., the FLEX series) offer a better chance of LFHP than perimodiolar arrays [7,25,26]; however, studies showing similar rates of LFHP between array types have also been published [19,27]. The present study can add little to this debate because: a) all the arrays used were lateral wall arrays; and b) comparison with the results of previous studies is difficult due to the differences inherent in comparing registry data to study data, e.g., exclusion/inclusion criteria and ways of reporting what constitutes successful LFHP. On the latter point, several studies have used the AAO scale (or something similar). Results featuring the slim modiolar array vary substantially: Iso-Mustajärvi et al. [28] reported that at ≥12 months postoperatively, 82.4 (14/17) participants had LFHP. Jimenez et al. [29] reported that 66.7% (10/15) participants had LFHP at a mean 4.6 months postoperatively. In contrast, Woodson et al. [30] found that 56% (22/39) of participants retained LFHP at activation and Kay-Rivest et al. [31] found that only 34.7% (16/46) of participants had LFHP at 1-year postoperative.

Helbig et al. [32] assessed LFHP using a variety of FLEX arrays (of which about half of which were FLEX20 or FLEX24 arrays and another quarter of which were older MED-EL arrays) and found that LFHP could be achieved in approximately 71.6% (58/81) of ears at 1 year postoperatively. Helbig et al. [18] investigated the FLEX20, FLEX24, FLEX28, and FLEX-SOFT, and while results appear to be impressive (see their Fig 5), they are not presented in such as way as to enable comparison. More recently, Moteki et al. [33] found 47.1% (8/17) of participants had LFHP at 6 months with the longer

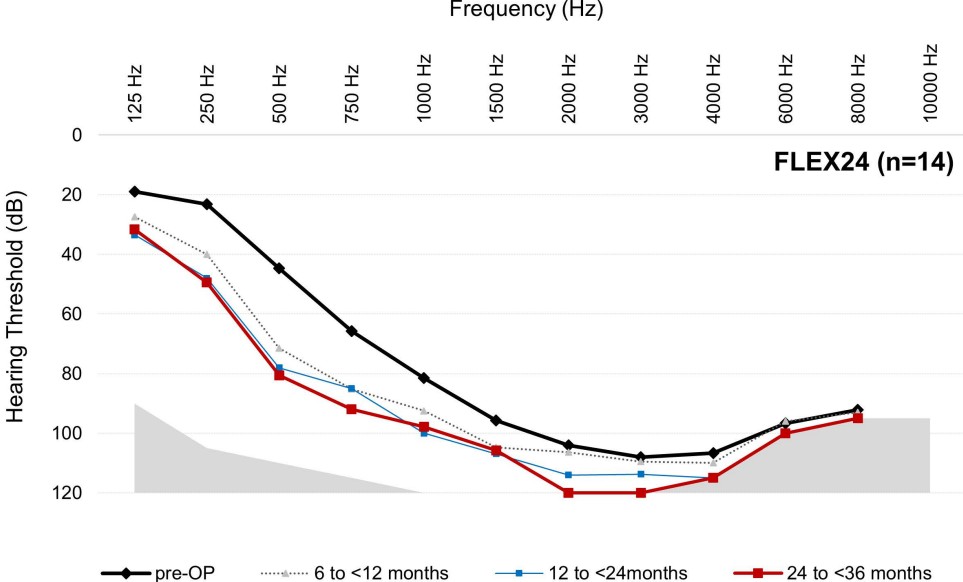

**Fig 3. Mean thresholds (dB HL) from 125 to 8000 Hz at each postoperative interval for the FLEX24 group.** Results are per ear (not per participant). Note, only 125–500 Hz are relevant for the AAO hearing loss assessment. n = number at preoperative cases. Number of postoperative audiogram data varies over time (see Table 3).

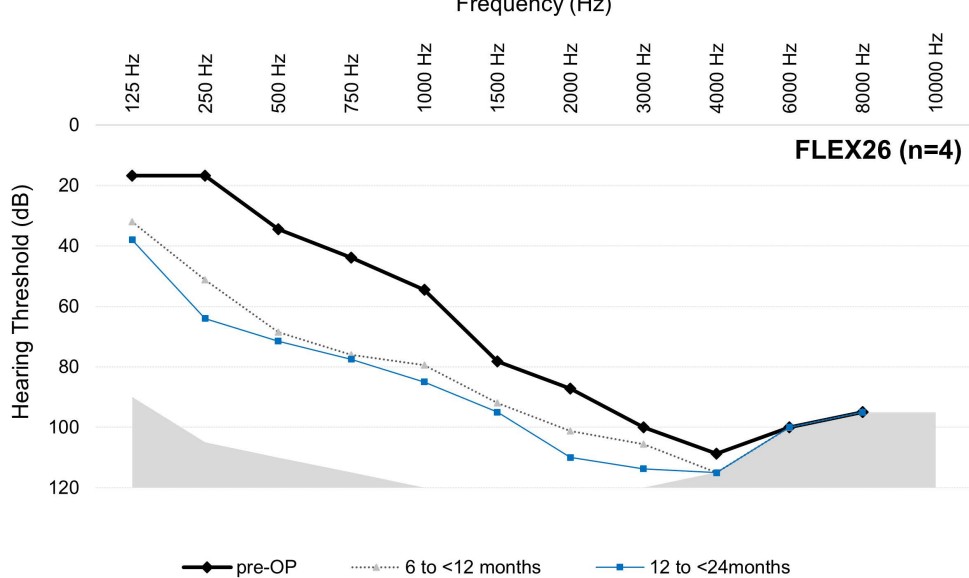

**Fig 4. Mean thresholds (dB HL) from 125 to 8000 Hz at each postoperative interval for the FLEX26 group.** Results are per ear (not per participant). Note, only 125–500 Hz are relevant for the AAO hearing loss assessment. n = number at preoperative cases. Number of postoperative audiogram data varies over time (see Table 3).

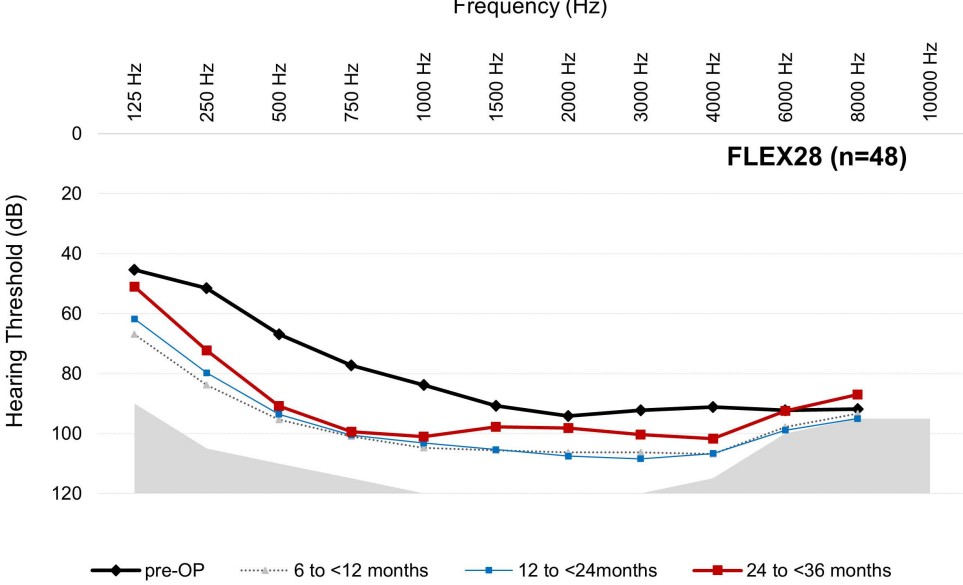

**Fig 5. Mean thresholds (dB HL) from 125 to 8000 Hz at each postoperative interval for FLEX28 group.** Results are per ear (not per participant). Note, only 125–500 Hz are relevant for the AAO hearing loss assessment. n = number at preoperative cases. Number of postoperative audiogram data varies over time (see Table 3).

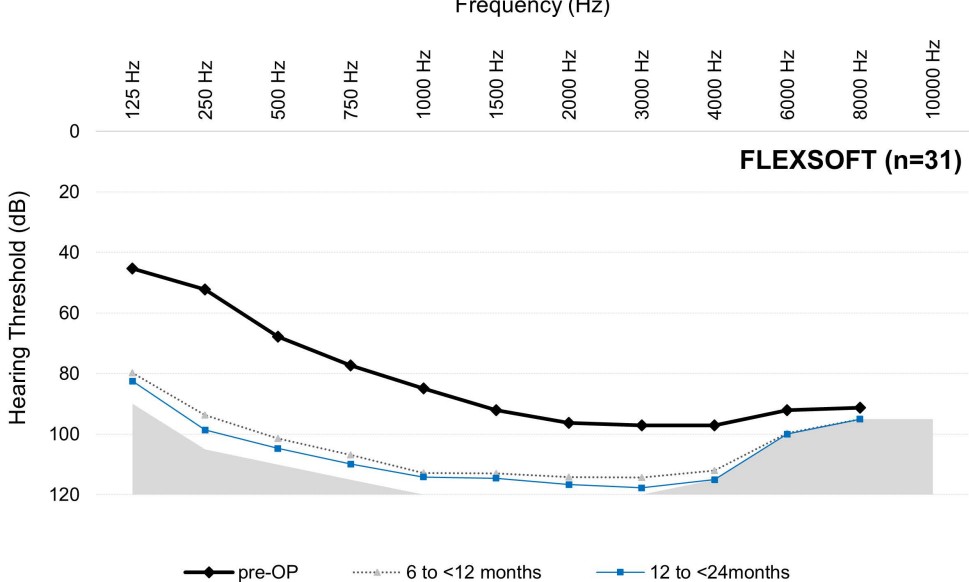

**Fig 6. Mean thresholds (dB HL) from 125 to 8000 Hz at each postoperative interval for the FLEXSOFT group.** Results are per ear (not per participant). Note, only 125–500 Hz are relevant for the AAO hearing loss assessment. n = number at preoperative cases. Number of postoperative audiogram data varies over time (see Table 3).

FLEX28 and FLEXSOFT arrays and Khan et al. [34] reported 78.4% (41/51) participants had LFHP at 1 year postoperatively (70.8% with manual insertion and 85.2% with robotic insertion) with FLEX20, FLEX24, and FLEX26 arrays. Results in the present study are worse than in Moteki et al. [33] but better than in Khan et al. [34], although, as mentioned previously, several factors make direct comparison problematic. Further, as per our study aims and data, while the results give a clear description of LFHP rates in real life in the included clinics within a certain timeframe, they are not of causal or explanatory value. To this end, multivariable modeling is theoretically possible; however, the current dataset and study design are not suited for robust causal inference, and future work should address this with prospective and hypothesis-driven analyses. In short, more research is necessary to determine the role that arrays and other factors play in LFHP.

The present study has some limitations. As stated previously, there is much more data for the FLEX28 and FLEXSOFT arrays than for the shorter FLEX24 and FLEX26 arrays. This is because the two longer arrays are more commonly used in participating clinics. An additional limitation is that the LFHP rate is difficult to track across intervals because of missing data (i.e., incomplete data sets) and the lower *n* at later intervals. This is probably due to bias: users may have opted to go to outpatient clinics and/or hearing centers for routine maintenance and only scheduled multiple follow-up appointments at the clinic when there were more pronounced hearing problems. Furthermore, not every clinic follows the same procedures, i.e., entering pre- and postoperative PTA. This is a typical aspect in working with real-word data generated from multiple clinics. Lastly, as stated earlier, various factors that (may) influence LFHP have not been controlled for.

As a counterpoint to these limitations, the MEHS registry contains sufficient detail to capture the use of the device, exposures, and outcomes of interest in the appropriate population. Patient selection and enrollment criteria minimized bias and ensured a representative real-world population. Our data support the claim that using FLEX electrode arrays positively contributes to increased levels of LFHP, particularly since our data includes CI users with a wide range of preoperative hearing thresholds, 4 centers with different peri-operative routines, many surgeons, and arrays of various lengths. This reduces the bias introduced by "surgeon factors" and "patient factors" on the overall effect on LFHP outcomes. The MEHS registry enables large volumes of real-world data to be investigated, which there is a known need for in working out the relative impact of different factors on LFHP [16]. To that end, we echo Sladen et al. [16] that future studies may find it beneficial to report patient population, inclusion/exclusion criteria, CI surgery technique applied (including insertion route of array), steroid/drug administration and dosage, array type and length used, follow-up period / outcome measure timing, and prognostic factors for LFHP studied. Adding more detail to studies, and by extension to the MEHS Registry, would be optimal. Until then, the present data allow us to state only that LFHP is possible, and sometimes probable, with FLEX arrays.

The clinical relevance of our findings is that LFHP for up to 3 years postoperatively is probable in many cases using FLEX series arrays. As regards the future, there are exciting developments that may make LFHP consistently more achievable, e.g., corticosteroid-eluting arrays [35,36], robotic array insertion [34,37], and a better matching of array size to candidates' morphologies.

## Conclusions

Low-frequency residual hearing preservation is possible with FLEX electrode arrays of all lengths for up to at least 3 years after implantation, even without accounting for factors known to contribute to successful hearing preservation. The MED-EL Hearing Solutions (MEHS) multicenter registry enables research with large amounts of real-life data collected from a variety of surgeons and hospitals during routine, everyday practice. Adding more factors to the data collected in the MEHS registry, like surgical techniques and the recipients' morphology, should increase its usefulness in assessing causality.

## Supporting information

**S1 Table. Recipients' pre- and post-operative hearing thresholds per frequency.** a) FLEX24 recipients b) FLEX26 recipients, c) FLEX28 recipients, d) FLEXSOFT recipients.
(DOCX)

 

**S2 Table. The number and percent of cases with matching or mismatching results with the VC and AAO classification systems.**
(DOCX)

## Acknowledgments

The authors would like to thank all the CI users who graciously agreed to contribute their data to the MEHS and Michael Todd (MED-EL) for writing and editing a version of this manuscript.

## Author contributions

**Conceptualization:** Ilona Anderson, Stefano Morettini, Vera Lohnherr, Daniel Polterauer-Neuling.

**Data curation:** Tobias Weissgerber, Magdalena Breu.

**Formal analysis:** Uwe Baumann, Karin A. Koinig.

**Investigation:** Uwe Baumann, Andreas Radeloff, Ilona Anderson, Stefano Morettini, Vera Lohnherr, Joachim Müller, Daniel Polterauer-Neuling.

**Methodology:** Tobias Weissgerber.

**Project administration:** Tobias Weissgerber, Daniel Polterauer-Neuling.

**Resources:** Tobias Weissgerber.

**Validation:** Tobias Weissgerber, Karin A. Koinig.

**Writing – review & editing:** Uwe Baumann, Tobias Weissgerber, Andreas Radeloff, Ilona Anderson, Karin A. Koinig, Magdalena Breu, Stefano Morettini, Vera Lohnherr, Joachim Müller, Daniel Polterauer-Neuling.

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
