## [Decision Letter · Decision Letter 0]

26 Sep 2025

Dear Dr. Polterauer,

Thank you for submitting your manuscript to PLOS ONE. After careful consideration, we feel that it has merit but does not fully meet PLOS ONE’s publication criteria as it currently stands. Therefore, we invite you to submit a revised version of the manuscript that addresses the points raised during the review process.

We look forward to receiving your revised manuscript.

Kind regards,

Toru Miwa

Academic Editor

PLOS ONE

Journal Requirements:

https://journals.plos.org/plosone/s/file?id=ba62/PLOSOne_formatting_sample_title_authors_affiliations.pdf....

“Funding from MED-EL. There is no grant number.”

Please include this amended Role of Funder statement in your cover letter; we will change the online submission form on your behalf

“Authors Ilona Anderson, Stefano Morettini, Karin Koinig, and Magdalena Breu are employed at MED-EL Elektromedizinische Geräte G.m.b.H.

Uwe Baumann and Daniel Polterauer have received travel support & research support from MED-EL GmbH (Austria).

Other authors disclose no conflict of interest.”

We note that one or more of the authors are employed by a commercial company: MED-EL GmbH

“This study was funded by MED-EL Elektromedizinische Geräte G.m.b.H. The funders assisted in study design, data collection and analysis, and preparation of the manuscript. Decision to publish was a joint decision between all authors”

“Funding from MED-EL. There is no grant number.”

Reviewers' comments:

Reviewer's Responses to Questions

**Comments to the Author**

1. Is the manuscript technically sound, and do the data support the conclusions?

Reviewer #1: Yes

Reviewer #2: Partly

2. Has the statistical analysis been performed appropriately and rigorously?

Reviewer #1: Yes

Reviewer #2: No

3. Have the authors made all data underlying the findings in their manuscript fully available?

Reviewer #1: Yes

Reviewer #2: Yes

4. Is the manuscript presented in an intelligible fashion and written in standard English?

Reviewer #1: Yes

Reviewer #2: Yes

Reviewer #1: Thank you for the opportunity to review this manuscript and for the effort invested in conducting the study and preparing the submission. As the authors mentioned, similar papers have been published; however, the data in this study were obtained from four hospitals in Germany, which helps reduce bias related to both surgeon and patient factors. Overall, I think the manuscript to be well written and informative.

Minor Points

1. In the legend of Figure 2, please remove the word “right.”

2. In Table 1, please change “gender” to “sex.”

Reviewer #2: This multicenter registry study examines low-frequency hearing preservation (LFHP) after cochlear implantation with MED-EL FLEX arrays (FLEX24/26/28/Soft) using the Vienna Consensus (VC) and AAO metrics at 6–36 months after surgery.

The paper shows LFHP is achievable across arrays, with generally higher rates for shorter arrays and notable divergence between VC and AAO results. The topic is clinically important and valuable, and with more robust statistical analysis the manuscript would be very suitable for publication.

Before specific points below, I recommend conducting formal statistical tests to evaluate overall/time and between-array differences and using multivariable models (with existing variables only) to identify drivers of LFHP (e.g., array length, baseline LF PTA, bilateral status, center, age/sex). Therefore, I recommend Major Revision.

Major concerns

1. Bilateral vs unilateral patients

A non-trivial proportion of participants are bilateral (VC 6.1%; AAO 11.6%), raising concerns that baseline hearing severity, neuroplasticity, and ear-to-ear dependencies could yield different LFHP trajectories than in unilateral cases. Simple ear-level analyses may underestimate uncertainty if left/right ears in bilateral recipients are treated as independent. Provide a subgroup comparison of unilateral vs bilateral recipients for key endpoints (VC categories; AAO <80 dB HL) with appropriate standard errors (e.g., cluster-robust by patient). For bilateral recipients, pre-specify a consistent ear-handling rule—such as analyzing the first implanted ear only as the primary approach or using a within-patient average where justified—and include a sensitivity analysis using the alternative rule to demonstrate robustness.

2. Uneven follow-up and informative missingness

Later windows have fewer cases and potential return-bias, complicating interpretation of cross-sectional proportions at 6–12/12–24/24–36 months. Fit a repeated-measures mixed-effects model for threshold change (VC bands) using all available visits. For AAO (<80 dB HL), add a time-to-event analysis (e.g., Kaplan–Meier to “loss of functional LFHP”) based on existing timestamps/interval bins, incorporating center fixed effects or a patient-level frailty term where feasible. These steps clarify temporal dynamics under missingness without requiring new data collection.

3. Quantify and interpret VC vs AAO discordance

FLEXSOFT shows moderate “preservation” by VC but low “functional preservation” by AAO—a clinically meaningful divergence rooted in different frequency ranges/thresholds. Where both metrics are available at the same visit, present a cross-tabulation (VC category × AAO LFHP yes/no) with agreement/discordance statistics (e.g., Cohen’s κ, McNemar’s test). Add a brief decision aid outlining when each metric should guide clinical choices (e.g., EAS candidacy versus electric-only).

Minor concerns

• Sample imbalance across arrays: Clearly note that FLEX28/FLEXSOFT cases dominate, reflecting practice but complicating direct comparisons across lengths; flag this in figure captions and in the Limitations.

• Safety summary placement: Reiterate in the Results that there was one reimplantation (cause unknown) and no device deficiencies, so safety can be grasped at a glance.

• Uncertainty and significance in figures/tables: For all plotted means (e.g., Figures 3–6), add SEs or 95% CIs, specify which is shown, and annotate the sample size (n) at each time point and per array. Where hypothesis tests are reported, state the test/model, whether two-sided, exact p-values, and effect sizes with 95% CIs; note any multiplicity handling.

• Define visit windows and pre-operative timing: Restate that pre-op unaided PTA was measured ≤12 months before surgery and that post-op windows are 6–12/12–24/24–36 months to aid reproducibility.

• Per-ear reporting clarity: Add “per ear” to figure/table titles and direct readers to the bilateral-handling rule in Methods.

.

Reviewer #1: **Yes:** Hidekane YoshimuraHidekane YoshimuraHidekane YoshimuraHidekane Yoshimura

Reviewer #2: No

---

## [Author Response · Author response to Decision Letter 1]

3 Nov 2025

EDITOR

and

The manuscript has been formatted to fit PLOS ONE.

“Funding from MED-EL. There is no grant number.”

Please include this amended Role of Funder statement in your cover letter; we will change the online submission form on your behalf

The cover letter has been amended to include this information.

“Authors Ilona Anderson, Stefano Morettini, Karin Koinig, and Magdalena Breu are employed at MED-EL Elektromedizinische Geräte G.m.b.H.

Uwe Baumann and Daniel Polterauer have received travel support & research support from MED-EL GmbH (Austria).

Other authors disclose no conflict of interest.”

We note that one or more of the authors are employed by a commercial company: MED-EL GmbH

The cover letter has been amended to include this information.

“This study was funded by MED-EL Elektromedizinische Geräte G.m.b.H. The funders assisted in study design, data collection and analysis, and preparation of the manuscript. Decision to publish was a joint decision between all authors”

“Funding from MED-EL. There is no grant number.”

This has been deleted from the paper & the cover letter has been amended to include this information.

It is nice to see this, so thank you. However, in this case it is not applicable.

There are a few changes we would like to inform you of.

1. We have updated Figures 1 and 2 to include an arrow that links the Sorting info at the top to the final n at the bottom, as illustrated in the screenshots below. This is the only change.

2. We have added a reference (#11) to a recently accepted paper on this registry. This provides context but does not change content. All reference #s after 11 have been updated accordingly.

3. The corresponding author’s last name is now “Polterauer-Neuling” and not Polterauer“

4. The affiliation for 5 has been edited, but it is the same place.

5. Please note, the authors’ ORCIDs are:

a. Uwe Baumann: 0000-0002-1295-2661

b. Andreas Radeloff: 0000-0003-1881-4179

c. Ilona Anderson: 0000-0001-7518-6661

d. Karin A Koinig: 0000-0002-3659-4934

e. Magdalena Breu: 0009-0007-4480-2098

f. Stefano Morettini: 0009-0007-1568-4993

g. Vera Lohnherr: 0000-0002-5655-7671

h. Joachim Müller: (no ORCID)

i. Daniel Polterauer-Neuling: 0000-0001-5008-7595

Reviewers' comments:

Reviewer's Responses to Questions

Comments to the Author

Reviewer #1: Thank you for the opportunity to review this manuscript and for the effort invested in conducting the study and preparing the submission. As the authors mentioned, similar papers have been published; however, the data in this study were obtained from four hospitals in Germany, which helps reduce bias related to both surgeon and patient factors. Overall, I think the manuscript to be well written and informative.

Thank you for the review & the kind words.

Minor Points

1. In the legend of Figure 2, please remove the word “right.”

Thank you, this has been corrected.

2. In Table 1, please change “gender” to “sex.”

Thank you, this has been corrected.

Reviewer #2: This multicenter registry study examines low-frequency hearing preservation (LFHP) after cochlear implantation with MED-EL FLEX arrays (FLEX24/26/28/Soft) using the Vienna Consensus (VC) and AAO metrics at 6–36 months after surgery.

The paper shows LFHP is achievable across arrays, with generally higher rates for shorter arrays and notable divergence between VC and AAO results. The topic is clinically important and valuable, and with more robust statistical analysis the manuscript would be very suitable for publication.

Before specific points below, I recommend conducting formal statistical tests to evaluate overall/time and between-array differences and using multivariable models (with existing variables only) to identify drivers of LFHP (e.g., array length, baseline LF PTA, bilateral status, center, age/sex). Therefore, I recommend Major Revision.

As stated in the manuscript, the primary aim of this data analyses was descriptive, that is to document the achievable rates of low-frequency hearing preservation (LFHP) across different MED-EL FLEX electrode arrays in a large, multicenter registry, and to compare outcomes according to two established hearing preservation definitions (Vienna Consensus and AAO). The analyses were not designed to explain longitudinal changes or to support causal inference regarding factors influencing LFHP. Accordingly, we did not conduct intra-individual longitudinal analyses, nor did we construct multivariable models for explanatory purposes.

Given that this is a registry study—with no obligation for CI users to return for follow-up visits, and with multiple outpatient facilities attended as alternative for follow up visits—we cannot ensure unbiased or continuous data collection. This would, however, be essential for robust intra-individual longitudinal analyses. Moreover, in the available registry data, variables such as electrode array length, baseline LF-PTA, and bilateral status are strongly interrelated. In addition, some key factors, such as course of deafness, hearing impairment duration, hearing aid usage prior to receiving an implant, or impedance measurements, were not included in this data set. This limits the ability to perform a multivariable model to isolate the independent effect of each variable, making robust causal or explanatory conclusions unreliable. Therefore, we believe that including such analyses would risk overstating the strength of any associations while missing out on some major factors affecting LFHP. We do not expect meaningful or clinically reliable correlations to emerge from such modeling.

Instead, the strength of our study is that it provides real-world evidence of LFHP across a spectrum of surgical practices and patient populations, while highlighting patterns of hearing preservation and differences between the Vienna Consensus and AAO definitions. We view this as a foundation on which future hypothesis-driven and longitudinal studies—designed explicitly to test causal relationships—can build.

That said, we took measures to strengthen the manuscript by:

1. Explicitly clarifying in the Introduction and Discussion that our objective was descriptive, not causal or explanatory: see lines: 83-84 and 285-290.

2. Reporting more detailed descriptive statistics in the appendix (e.g., group-level differences between arrays and over time) to aid interpretation while maintaining the descriptive scope: for see the appendix tables.

3. Highlighting in the Discussion that while multivariable modeling is theoretically possible, the current dataset and study design are not suited for robust causal inference, and that future work should address this with prospective and hypothesis-driven analyses: see lines: 291-295.

We hope this adequately addresses the reviewer’s concern while maintaining the integrity of our study design and objectives.

Major concerns

1. Bilateral vs unilateral patients

A non-trivial proportion of participants are bilateral (VC 6.1%; AAO 11.6%), raising concerns that baseline hearing severity, neuroplasticity, and ear-to-ear dependencies could yield different LFHP trajectories than in unilateral cases. Simple ear-level analyses may underestimate uncertainty if left/right ears in bilateral recipients are treated as independent. Provide a subgroup comparison of unilateral vs bilateral recipients for key endpoints (VC categories; AAO <80 dB HL) with appropriate standard errors (e.g., cluster-robust by patient). For bilateral recipients, pre-specify a consistent ear-handling rule—such as analyzing the first implanted ear only as the primary approach or using a within-patient average where justified—and include a sensitivity analysis using the alternative rule to demonstrate robustness.

Our analytic intention, however, was not to model dependencies or to draw causal inferences about trajectories but to provide descriptive, ear-level summaries of LFHP outcomes across a broad, real-world registry. We chose ear-level as the unit of analysis because:

1. LFHP is an ear-specific phenomenon, influenced by surgical approach and array choice in that ear

2. The registry data are structured at the ear level, with no prespecified analytic framework for handling bilateral cases. By design we only know whether a patient is bilaterally implanted if CIs are included for both ears (thus, if the other ear had a non-MED-EL device or was implanted in a clinic that does not participate in the registry, we would not know that the person is a bilateral CI user),

3. From what we do know, the proportion of bilateral recipients is relatively modest (6.1% VC; 11.6% AAO)

We consider therefore our main findings as not meaningfully altered by bilateral status or analytic choice, while remaining aligned with the descriptive scope of our study.

2. Uneven follow-up and informative missingness

Later windows have fewer cases and potential return-bias, complicating interpretation of cross-sectional proportions at 6–12/12–24/24–36 months. Fit a repeated-measures mixed-effects model for threshold change (VC bands) using all available visits. For AAO (<80 dB HL), add a time-to-event analysis (e.g., Kaplan–Meier to “loss of functional LFHP”) based on existing timestamps/interval bins, incorporating center fixed effects or a patient-level frailty term where feasible. These steps clarify temporal dynamics under missingness without requiring new data collection.

This is a registry study and, as such, looks at real world data to provide real world evidence. As a limiting consequence, participant numbers vary across follow-up intervals.

This reflects typical clinical practice: some patients do not come back, especially if they experience no issues. Other patients opt of follow ups at trained hearing aid centres instead of the participating clinics, as this can be more convenient in Germany.

3. Quantify and interpret VC vs AAO discordance

FLEXSOFT shows moderate “preservation” by VC but low “functional preservation” by AAO—a clinically meaningful divergence rooted in different frequency ranges/thresholds. Where both metrics are available at the same visit, present a cross-tabulation (VC category × AAO LFHP yes/no) with agreement/discordance statistics (e.g., Cohen’s κ, McNemar’s test). Add a brief decision aid outlining when each metric should guide clinical choices (e.g., EAS candidacy versus electric-only).

Thanks for noticing that FLEXSOFT illustrates differing results when applying the Vienna Consensus (VC) versus AAO metrics. However, we would like to clarify that this divergence is not clinically meaningful but rather an intrinsic feature of the two scales, which were developed to capture different aspects of hearing preservation (HP). The VC categorizes relative hearing preservation at 250, 500, and 1000 1500 Hz based on postoperative threshold shifts across a broad frequency range. Its purpose is to enable standardized, comparable reporting across centers and studies, irrespective of whether the absolute thresholds remain within a functionally usable range. The AAO metric, by contrast, is a clinically oriented framework that defines preservation as maintaining ≤80 dB HL at 125, 250, and 500 Hz—frequencies that are critical for electro-acoustic stimulation (EAS). It therefore directly addresses whether residual hearing remains of functional benefit.

Therefore, because the VC and AAO frameworks were designed to answer different questions, and because they assess different frequencies, agreement or discordance between them is neither expected nor clinically meaningful. A cross-tabulation or concordance analysis would risk suggesting comparability between two scales that are not intended to be interchangeable. Instead, their divergence simply reflects their different conceptual focus: VC quantifies relative preservation, whereas AAO defines functional preservation. (Note: This is now stated in the Methods and we have changed the 1st word of the paper’s title from “Functional” to “Low frequency”.)

This is also stressed by our results for the FLEXSOFT array—the longest in o

---

## [Decision Letter · Decision Letter 1]

19 Dec 2025

Dear Dr. Polterauer-Neuling,

Thank you for submitting your manuscript to PLOS ONE. After careful consideration, we feel that it has merit but does not fully meet PLOS ONE’s publication criteria as it currently stands. Therefore, we invite you to submit a revised version of the manuscript that addresses the points raised during the review process.

We look forward to receiving your revised manuscript.

Kind regards,

Toru Miwa

Academic Editor

PLOS One

Journal Requirements:

Reviewer's Responses to Questions

**Comments to the Author**

Reviewer #2: All comments have been addressed

Reviewer #3: (No Response)

Reviewer #4: All comments have been addressed

Reviewer #5: All comments have been addressed

Reviewer #6: (No Response)

Reviewer #7: (No Response)

2. Is the manuscript technically sound, and do the data support the conclusions?

Reviewer #2: Partly

Reviewer #3: Partly

Reviewer #4: Yes

Reviewer #5: Yes

Reviewer #6: Yes

Reviewer #7: Partly

3. Has the statistical analysis been performed appropriately and rigorously?

Reviewer #2: No

Reviewer #3: Yes

Reviewer #4: Yes

Reviewer #5: Yes

Reviewer #6: Yes

Reviewer #7: N/A

4. Have the authors made all data underlying the findings in their manuscript fully available?

Reviewer #2: Yes

Reviewer #3: Yes

Reviewer #4: Yes

Reviewer #5: Yes

Reviewer #6: Yes

Reviewer #7: Yes

5. Is the manuscript presented in an intelligible fashion and written in standard English?

Reviewer #2: Yes

Reviewer #3: Yes

Reviewer #4: Yes

Reviewer #5: Yes

Reviewer #6: Yes

Reviewer #7: Yes

Reviewer #2: Thank you for the opportunity to review the revised version of this manuscript entitled “Low frequency hearing preservation after cochlear implant surgery is achievable with each FLEX electrode array: Real-world evidence from the MEHS Registry.”

The authors have addressed the previous comments with clarity and appropriate modifications. The objective of the study as descriptive has been clearly restated, and additional context has been added to improve interpretation. The manuscript provides clinically meaningful real-world data, and the overall structure is sound.

However, there remain a few points that should be clarified further to ensure appropriate interpretation of the findings, particularly regarding the limitations of the descriptive approach given sample size imbalances. I recommend Minor Revision.

Major Points

1)Limitations of the descriptive approach due to subgroup size and imbalance

While the authors correctly emphasize that the study was designed as descriptive, this methodological approach is most informative when subgroup sample sizes are sufficiently large and balanced. In the present analysis, some subgroups—particularly FLEX26 and FLEXSOFT at later time intervals—include only a small number of cases.

Please clearly acknowledge in the Discussion that the interpretive strength of subgroup trends is limited due to small and uneven sample sizes, and that these results should be interpreted with caution.

2) Avoiding causal interpretation in observational statements

Some wording in the Discussion still implies causality (e.g., privilege of shorter arrays in achieving better LFHP). Please rephrase such statements to indicate observational tendencies only (e.g., “a trend was observed that…” or “in this descriptive dataset, shorter arrays tended to show…”).

Reviewer #3: Dear

Thank You for manuscript . After previous review manuscript looks much better , but still there are several flaws. In introduction i can t see data or background and "fascinating" database . Register could be beneficial but better will be to to link that to essence of the study. Lenhardt "soft " technique had not so much connected with hearing preservation so that part of the introduction should be significantly adjusted to real history of that .

In results there is a lot of nice analysis but honestly there is significant decrease of the hearing thresholds so it should be the main issue why hearing was not preserved . Also after check there are other types of evaluation of that preservation and if applied VC it should be correlated with other more common and validated tools for that .

Reviewer #4: This is a well-written descriptive paper of hearing preservation defined by two classification systems using multiple different electrode array styles. The authors do well to temper their inferences and conclusions within the context of a descriptive database study, for which multiple factors which are thought to contribute to LFHP cannot be controlled in a prospective fashion, but can be appropriately described. This study adds valuable insights to the field, notably with the inclusion of hearing preservation rates within the FLEXSOFT group. One notable strength of the study is the inclusion of multiple surgeons and centers from which the data was drawn from. Results are appropriately compared to existing literature.

The Authors have provided a comprehensive response to reviewer critiques. Notably, I fully support the author's response to reviewer 2 critiques and agree with the authors in their approach to data analysis (mainly that this is a descriptive study, not meant to make statistically backed conclusions on factors influencing HP (such as electrode style)). Additionally, I agree with their approach to handling of individual ears in the setting of bilateral implantation. These efforts from the Authors appropriately temper their conclusions from the study within the bounds of its design.

Reviewer #5: (No Response)

Reviewer #6: PONE-D-25-39581_R1: Low frequency hearing preservation after cochlear implant surgery is achievable with each FLEX electrode array: Real world evidence from the MEHS Registry.

This paper presents the results of a registry study investigating the preservation of residual hearing with different cochlear implant (CI) electrodes, classified using two different methods. The strength of this approach lies in the large number of patients.

I was apparently brought in as an additional reviewer because one of the two original reviewers and the authors could not reach an agreement. Overall, the manuscript is well-written and presents relevant data. Nevertheless, I have a point of my own, which I believe needs to be added, before addressing the points of contention.

In my opinion, the statistics requested by Reviewer 2 make little sense. But what I find completely lacking is a contextualization of the results within the current literature, specifically the achieved residual hearing preservation rates for each electrode length. There are many studies where individual clinics investigate residual hearing preservation under more consistent conditions, naturally with significantly fewer patients each. Here, the rates from this large registry study need to be compared with the data from the literature.

a)

Reviewer 2:

“Before specific points below, I recommend conducting formal statistical tests to evaluate overall/time and between-array differences and using multivariable models (with existing variables only) to identify drivers of LFHP (e.g., array length, baseline LF PTA, bilateral status, center, age/sex). Therefore, I recommend Major Revision.”

Author’s response:

“Given that this is a registry study—with no obligation for CI users to return for follow-up visits, and with multiple outpatient facilities attended as alternative for follow up visits—we cannot ensure unbiased or continuous data collection. This would, however, be essential for robust intraindividual longitudinal analyses. Moreover, in the available registry data, variables such as electrode array length, baseline LF-PTA, and bilateral status are strongly interrelated. In addition, some key factors, such as course of deafness, hearing impairment duration, hearing aid usage prior to receiving an implant, or impedance measurements, were not included in this data set. This limits the ability to perform a multivariable model to isolate the independent effect of each variable, making robust causal or explanatory conclusions unreliable. Therefore, we believe that including such analyses would risk overstating the strength of any associations while missing out on some major factors affecting LFHP. We do not expect meaningful or clinically reliable correlations to emerge from such modeling.”

My assessment:

I agree with the authors. It is very regrettable that the dataset is not more comprehensive and has many gaps. However, this is not a prospective study with precisely controlled conditions, but rather a retrospective analysis of existing data from various clinics, which, based on experience, can vary considerably in detail. For example, I am surprised that the preoperative audiogram is up to twelve months old in some cases, and this heavily reduces the significance of the preoperative residual hearing. According to extensive literature, the duration and progression of hearing loss, hearing aid use, and deafness also have a considerable influence on the preservation of residual hearing. If this data is simply not available, it makes no sense, in my view, to feign accuracy through extensive statistics.

b)

Reviewer 2:

“A non-trivial proportion of participants are bilateral (VC 6.1%; AAO 11.6%), raising concerns that baseline hearing severity, neuroplasticity, and ear-to-ear dependencies could yield different LFHP trajectories than in unilateral cases. Simple ear-level analyses may underestimate uncertainty if left/right ears in bilateral recipients are treated as independent. Provide a subgroup comparison of unilateral vs bilateral recipients for key endpoints (VC categories; AAO <80 dB HL) with appropriate standard errors (e.g., cluster-robust by patient). For bilateral recipients, pre-specify a consistent ear-handling rule—such as analyzing the first implanted ear only as the primary approach or using a within-patient average where justified—and include a sensitivity analysis using the alternative rule to demonstrate robustness.

Author’s response:

“Our analytic intention, however, was not to model dependencies or to draw causal inferences about trajectories but to provide descriptive, ear-level summaries of LFHP outcomes across a broad, real-world registry. We chose ear-level as the unit of analysis because:

1. LFHP is an ear-specific phenomenon, influenced by surgical approach and array choice in that ear

2. The registry data are structured at the ear level, with no prespecified analytic framework

for handling bilateral cases. By design we only know whether a patient is bilaterally

implanted if CIs are included for both ears (thus, if the other ear had a non-MED-EL device

or was implanted in a clinic that does not participate in the registry, we would not know

that the person is a bilateral CI user),

3. From what we do know, the proportion of bilateral recipients is relatively modest (6.1%

VC; 11.6% AAO)

We consider therefore our main findings as not meaningfully altered by bilateral status or

analytic choice, while remaining aligned with the descriptive scope of our study.”

My assessment:

I agree with the authors, but I have a suggestion for further analysis, though probably only in the future when more bilateral datasets are available. The surgeon only manipulates the peripheral auditory system. Effects resulting from bilateral connections are not to be expected. The authors rightly argue that the necessary data is not available with the required clarity. This is also because it would be necessary to consider not only a cochlear implant (CI) on the contralateral side, but also residual hearing and/or hearing aid use. This data would be essential for any analysis.

Nevertheless, I find another effect very interesting. To investigate this, there are probably not yet enough datasets. If a CI is implanted on the second side, is the preservation of residual hearing comparable to that on the first side, or not? In other words, if the residual hearing on the first side is well preserved, does the patient then have a better chance on the second side? And if residual hearing is lost on the first side and no unusual occurrence is noted intraoperatively (which is naturally not recorded in the registry), does the patient then also have an increased risk on the second side? Because it may also be that factors such as predisposition or the nature of the underlying disease play a role, which have so far been recorded very little.

c)

Reviewer 2:

“Later windows have fewer cases and potential return-bias, complicating interpretation of crosssectional proportions at 6–12/12–24/24–36 months. Fit a repeated-measures mixed-effects model for threshold change (VC bands) using all available visits. For AAO (<80 dB HL), add a timeto- event analysis (e.g., Kaplan–Meier to “loss of functional LFHP”) based on existing timestamps/interval bins, incorporating center fixed effects or a patient-level frailty term where feasible. These steps clarify temporal dynamics under missingness without requiring new data collection.

Author’s response:

“This is a registry study and, as such, looks at real world data to provide real world evidence. As a limiting consequence, participant numbers vary across follow-up intervals.

This reflects typical clinical practice: some patients do not come back, especially if they

experience no issues. Other patients opt of follow ups at trained hearing aid centres instead of the participating clinics, as this can be more convenient in Germany.”

My assessment:

I agree with the authors. In my opinion, the purpose of the study did not require these statistics; rather, these statistics would give a false impression of accuracy.

d)

Reviewer 2:

“FLEXSOFT shows moderate “preservation” by VC but low “functional preservation” by AAO—a clinically meaningful divergence rooted in different frequency ranges/thresholds. Where both metrics are available at the same visit, present a cross-tabulation (VC category × AAO LFHP yes/no) with agreement/discordance statistics (e.g., Cohen’s κ, McNemar’s test). Add a brief decision aid outlining when each metric should guide clinical choices (e.g., EAS candidacy versus electric-only).“

Author’s response:

“Thanks for noticing that FLEXSOFT illustrates differing results when applying the Vienna

Consensus (VC) versus AAO metrics. However, we would like to clarify that this divergence is not clinically meaningful but rather an intrinsic feature of the two scales, which were developed to capture different aspects of hearing preservation (HP). The VC categorizes relative hearing preservation at 250, 500, and 1000 1500 Hz based on postoperative threshold shifts across a broad frequency range. Its purpose is to enable standardized, comparable reporting across centers and studies, irrespective of whether the absolute thresholds remain within a functionally usable range. The AAO metric, by contrast, is a clinically oriented framework that defines preservation as maintaining ≤80 dB HL at 125, 250, and 500 Hz—frequencies that are critical for electro-acoustic stimulation (EAS). It therefore directly addresses whether residual hearing remains of functional benefit.

Therefore, because the VC and AAO frameworks were designed to answer different questions, and because they assess different frequencies, agreement or discordance between them is neither expected nor clinically meaningful. A cross-tabulation or concordance analysis would risk suggesting comparability between two scales that are not intended to be interchangeable. Instead, their divergence simply reflects their different conceptual focus: VC quantifies relative preservation, whereas AAO defines functional preservation. (Note: This is now stated in the Methods and we have changed the 1st word of the paper’s title from “Functional” to “Low frequency”.)

This is also stressed by our results for the FLEXSOFT array—the longest in our cohort and

therefore potentially more traumatic than the shorter variants—it is expected that preservation may appear moderate under VC (reflecting limited threshold shift), while AAO indicates poor functional preservation if residual thresholds exceed 80 dB HL at low frequencies. This is not a contradiction but a natural consequence of applying two distinct scales which indeed capture different aspects of HP in a cohort of FLEXSOFT candidates who typically lack low-frequency residual hearing even pre-operatively.

In summary, the observed divergence between VC and AAO outcomes is not a clinically

meaningful inconsistency but an expected reflection of the complementary purposes of these two scales”

My assessment:

Here I find myself caught between reviewer and authors. Since both metrics (VC and AAO) have different objectives, statistics are of little use. However, I too would appreciate a descriptive comparison showing in which and how many cases the results of both metrics coincide and diverge; a table would suffice for this. The discussion should then include a paragraph that elaborates on the differences and provides an indication of which metric is better suited for which application.

Reviewer #7: I appreciate the opportunity to review the manuscript “Low frequency hearing preservation after cochlear implant surgery is achievable with each FLEX electrode array: Real world evidence from the MEHS Registry”. It is definitely still important to evaluate hearing preservation (HP) after CI surgery. The rationale for the study is certainly warranted and interesting aspect is to use real world evidence from the MEHS Registry. However, There are several major concerns that currently make the manuscript unsuitable for publication: 1) methodological unresolved dilemmas; 2) difficulties in interpreting between-electrode differences in hearing preservation (HP); 3) limited novelty of the study

1. Methodological concerns

A substantial number of previous studies have investigated hearing-preservation (HP) width using various approaches. At present, two systems are most commonly used: the HearRing Group Classification System (Skarzynski et al., 2013) and the American Academy of Otolaryngology (AAO) reporting system. The HearRing system, in particular, has been widely adopted and cited more than 200 times. It is therefore unclear why the authors chose to introduce yet another matrix for HP evaluation (Vienna Consensus- VC). By employing a new, non-standard scale, the opportunity to compare the findings with previously published results is significantly impaired. Furthermore, the manuscript does not reference other applications of the VC method, and it appears that VC has not yet been established as a validated assessment tool. As HP evaluation is described in the manuscript, there is a critical methodological dilemma regarding how to treat unmeasurable thresholds. The authors state:

“We truncated all measurements exceeding the maximum audiometer test thresholds to the values given in Table IV in Skarzynski et al., to obtain a uniform maximum per frequency.”

These maximum values are 105 dB, 110 dB and 120 dB for 250 Hz, 500 Hz and 1000 Hz respectively, resulting in a PTLF of 111.67 dB.

According to the authors:

“To be assessed via VC, CI recipients must have some preoperative hearing at those frequencies, i.e., ≤ 111.67 dB HL (average maximum output levels).”

However, this leads to conceptual inconsistencies. For example, if a patient has a preoperative threshold at the maximum output levels and postoperatively loses all measurable hearing (i.e., no auditory sensation), the VC method may still classify this case as complete preservation, because both values reside at or beyond the truncated ceiling. Similarly, a patient losing hearing at two frequencies but retaining ≤15 dB preservation at one frequency might also be misclassified as full preservation. Such outcomes generate bias and undermine interpretability. Therefore, I request clarification on how the authors handled unmeasurable thresholds (no hearing at all) within the VC framework. In addition, I strongly recommend applying the HearRing Classification System, which was explicitly designed to avoid this dilemma. The HearRing formula for qualitative HP classification:

Relative change = ((PTApost-PTApre)/(PTAmax-PTApre))

Where:

PTApost is pure tone average measured postoperatively; PTApre is pure tone average measured pre-operatively; PTAmax is the limits of the audiometer

This formula effectively resolves the bias introduced by unmeasurable thresholds. Because the classification scales results relative to the patient’s preoperative hearing, it prevents worse preoperative hearing from artificially inflating postoperative HP outcomes. The equation presents the relative change, as a percentage of hearing loss, Than, the relative change is converted to preservation by calculating 100% - relative change in %. Then, a percentage of hearing preserved can be converted into a categorical scale (complete, partial, minimal, or loss). This approach can also be applied specifically to the three low frequencies. I recommend recalculating HP rates using the HearRing Classification System.

2. Interpretation of electrode-related differences

The authors suggest that low-frequency hearing preservation (LFHP) appears more likely when shorter arrays are used because these arrays (e.g., FLEX24) may not reach apical low-frequency regions and therefore may not induce trauma or inflammation in these areas. However, this interpretation is not adequately justified given the methodology.

Several key confounding factors—surgical technique, patient age at implantation, onset and progression of hearing loss—were not controlled. Additionally, the preoperative thresholds differ significantly between electrode groups; notably, the FLEX24 group had better preoperative hearing. There is substantial evidence that preoperative thresholds predict postoperative HP outcomes (Lee et al., 2020; Wanna et al., 2017). Therefore, attributing differences solely to electrode length is not warranted without controlling for these variables.

3. Limited novelty

The conclusion that low-frequency residual hearing can be preserved with FLEX electrodes of all lengths is already well established in the literature. For example, the systematic review by Van de Heyning et al. (2022), “Systematic Literature Review of Hearing Preservation Rates in Cochlear Implantation Associated With Medium- and Longer-Length Flexible Lateral Wall Electrode Arrays”, comprehensively documents this. The present manuscript does not provide sufficient new insight beyond existing knowledge.

.

Reviewer #2: No

Reviewer #3: No

Reviewer #4: **Yes:** Alexander Dale ClaussenAlexander Dale ClaussenAlexander Dale ClaussenAlexander Dale Claussen

Reviewer #5: No

Reviewer #6: No

Reviewer #7: No

---

## [Author Response · Author response to Decision Letter 2]

22 Jan 2026

For all reviewers:

Please note that we have slightly changed the title: 1) we added “functional” to account for the AAO results; 2) we deleted “each” because there exist FLEX arrays (e.g., FLEX20, FLEX34) that are not featured in this study, thus “each” is incorrect.

We have also made some edits to clarify that one scale is hearing preservation and the other is functional hearing preservation, e.g., line 198.

Reviewer #2: Thank you for the opportunity to review the revised version of this manuscript entitled “Low frequency hearing preservation after cochlear implant surgery is achievable with each FLEX electrode array: Real-world evidence from the MEHS Registry.”

The authors have addressed the previous comments with clarity and appropriate modifications. The objective of the study as descriptive has been clearly restated, and additional context has been added to improve interpretation. The manuscript provides clinically meaningful real-world data, and the overall structure is sound.

However, there remain a few points that should be clarified further to ensure appropriate interpretation of the findings, particularly regarding the limitations of the descriptive approach given sample size imbalances. I recommend Minor Revision.

Major Points

1)Limitations of the descriptive approach due to subgroup size and imbalance

While the authors correctly emphasize that the study was designed as descriptive, this methodological approach is most informative when subgroup sample sizes are sufficiently large and balanced. In the present analysis, some subgroups—particularly FLEX26 and FLEXSOFT at later time intervals—include only a small number of cases.

Please clearly acknowledge in the Discussion that the interpretive strength of subgroup trends is limited due to small and uneven sample sizes, and that these results should be interpreted with caution. Agreed. We’ve reworked the paragraph from 266-88 to make this more explicit.

2) Avoiding causal interpretation in observational statements

Some wording in the Discussion still implies causality (e.g., privilege of shorter arrays in achieving better LFHP). Please rephrase such statements to indicate observational tendencies only (e.g., “a trend was observed that…” or “in this descriptive dataset, shorter arrays tended to show…”). Excellent point. Please see the response to the comment above, lines 34-6 (deletion) in the plain language summary, lines 212-3, and lines 318-324.

Reviewer #3: Dear

Thank You for manuscript . After previous review manuscript looks much better , but still there are several flaws. In introduction i can t see data or background and "fascinating" database We are not sure what this means. The place for data is the Results. The MEHS is covered in the Methods and Materials. Register could be beneficial but better will be to to link that to essence of the study. Thispaper is based on a registry, with a protocol, which makes it rigorous. Lenhardt "soft " technique had not so much connected with hearing preservation so that part of the introduction should be significantly adjusted to real history of that . As is well-known, soft surgery is a key component of HP surgery. Lenhardt’s paper is the essence of soft surgery, and many papers reference it.

In results there is a lot of nice analysis but honestly there is significant decrease of the hearing thresholds so it should be the main issue why hearing was not preserved . This is as to be expected in CI. As this was an unbiased study using real-world data from a registry where all patients were analysed, there was no control for surgical technique, or experience of surgeon BUT STILL hearing was preserved. Also after check there are other types of evaluation of that preservation and if applied VC it should be correlated with other more common and validated tools for that . AAO was also used, so two measurements, one looking at hearing preservation and the other looking at functional hearing preservation.

Reviewer #4: This is a well-written descriptive paper of hearing preservation defined by two classification systems using multiple different electrode array styles. The authors do well to temper their inferences and conclusions within the context of a descriptive database study, for which multiple factors which are thought to contribute to LFHP cannot be controlled in a prospective fashion, but can be appropriately described. This study adds valuable insights to the field, notably with the inclusion of hearing preservation rates within the FLEXSOFT group. One notable strength of the study is the inclusion of multiple surgeons and centers from which the data was drawn from. Results are appropriately compared to existing literature.

The Authors have provided a comprehensive response to reviewer critiques. Notably, I fully support the author's response to reviewer 2 critiques and agree with the authors in their approach to data analysis (mainly that this is a descriptive study, not meant to make statistically backed conclusions on factors influencing HP (such as electrode style)). Additionally, I agree with their approach to handling of individual ears in the setting of bilateral implantation. These efforts from the Authors appropriately temper their conclusions from the study within the bounds of its design. Thank you

Reviewer #6: PONE-D-25-39581_R1: Low frequency hearing preservation after cochlear implant surgery is achievable with each FLEX electrode array: Real world evidence from the MEHS Registry.

This paper presents the results of a registry study investigating the preservation of residual hearing with different cochlear implant (CI) electrodes, classified using two different methods. The strength of this approach lies in the large number of patients.

I was apparently brought in as an additional reviewer because one of the two original reviewers and the authors could not reach an agreement. Overall, the manuscript is well-written and presents relevant data. Nevertheless, I have a point of my own, which I believe needs to be added, before addressing the points of contention.

In my opinion, the statistics requested by Reviewer 2 make little sense. But what I find completely lacking is a contextualization of the results within the current literature, specifically the achieved residual hearing preservation rates for each electrode length. There are many studies where individual clinics investigate residual hearing preservation under more consistent conditions, naturally with significantly fewer patients each. Here, the rates from this large registry study need to be compared with the data from the literature.

We understand what you mean, however all other studies already have the inclusion criteria of participants with substantial pre-op residual hearing & the use of soft-surgery techniques to preserve the residual hearing. That is to say, the results they present show what is possible, not what actually goes in the real world. The present study is unbiased in that it uses real-world data from a registry where all patients were analysed: there was no control for surgical technique or experience of surgeon and thus cannot reasonably be compared to other studies. This would be like comparing apples to oranges.

a)

Reviewer 2:

“Before specific points below, I recommend conducting formal statistical tests to evaluate overall/time and between-array differences and using multivariable models (with existing variables only) to identify drivers of LFHP (e.g., array length, baseline LF PTA, bilateral status, center, age/sex). Therefore, I recommend Major Revision.”

Author’s response:

“Given that this is a registry study—with no obligation for CI users to return for follow-up visits, and with multiple outpatient facilities attended as alternative for follow up visits—we cannot ensure unbiased or continuous data collection. This would, however, be essential for robust intraindividual longitudinal analyses. Moreover, in the available registry data, variables such as electrode array length, baseline LF-PTA, and bilateral status are strongly interrelated. In addition, some key factors, such as course of deafness, hearing impairment duration, hearing aid usage prior to receiving an implant, or impedance measurements, were not included in this data set. This limits the ability to perform a multivariable model to isolate the independent effect of each variable, making robust causal or explanatory conclusions unreliable. Therefore, we believe that including such analyses would risk overstating the strength of any associations while missing out on some major factors affecting LFHP. We do not expect meaningful or clinically reliable correlations to emerge from such modeling.”

My assessment:

I agree with the authors. It is very regrettable that the dataset is not more comprehensive and has many gaps. However, this is not a prospective study with precisely controlled conditions, but rather a retrospective analysis of existing data from various clinics, which, based on experience, can vary considerably in detail. For example, I am surprised that the preoperative audiogram is up to twelve months old in some cases, and this heavily reduces the significance of the preoperative residual hearing. According to extensive literature, the duration and progression of hearing loss, hearing aid use, and deafness also have a considerable influence on the preservation of residual hearing. If this data is simply not available, it makes no sense, in my view, to feign accuracy through extensive statistics.

Thank you and yes, real-world data shows the reality of clinical procedure. While some gaps in datasets of real-world evidence are unavoidable, we are working to improve the registry to fill-in these gaps where we can.

We looked at the data again and found that more than 75% of the pre-op assessments took place within 3 months of implantation (and 62% were within 10 days of implantation). We’ve added some sentences to this end (see lines 199-202). On the point, we’d like to point out that using older pre-op assessments would tend to underestimate the HP because while it’s quite possibly that candidates’ PTA is worse or equal at 10 days pre-op than it was at 11 months pre-op, but it’s very unlikely to have improved from 11m to 10d. Therefore, if a better PTA from an earlier assessment is used as the baseline for comparison with the post-operative PTA, this would lead to an underestimation of hearing preservation rather than an overestimation.

We agree that we would not want to extend the statistics further based on the limitations of pre-op information (please see our original answer to the reviewer from the last round: “some key factors, such as course of deafness, hearing impairment duration, hearing aid usage prior to receiving an implant, or impedance measurements, were not included in this data set”…). And, in the Discussion, we catalogue the potential casual factors for hearing preservation that we don’t know. We can’t present a nuanced picture here; however, the data do allow us to substantiate the claim that regardless of all the factors that may or may not influence HP, complete or partial HP was possible with FLEX electrode arrays.

b)

Reviewer 2:

“A non-trivial proportion of participants are bilateral (VC 6.1%; AAO 11.6%), raising concerns that baseline hearing severity, neuroplasticity, and ear-to-ear dependencies could yield different LFHP trajectories than in unilateral cases. Simple ear-level analyses may underestimate uncertainty if left/right ears in bilateral recipients are treated as independent. Provide a subgroup comparison of unilateral vs bilateral recipients for key endpoints (VC categories; AAO <80 dB HL) with appropriate standard errors (e.g., cluster-robust by patient). For bilateral recipients, pre-specify a consistent ear-handling rule—such as analyzing the first implanted ear only as the primary approach or using a within-patient average where justified—and include a sensitivity analysis using the alternative rule to demonstrate robustness.

Author’s response:

“Our analytic intention, however, was not to model dependencies or to draw causal inferences about trajectories but to provide descriptive, ear-level summaries of LFHP outcomes across a broad, real-world registry. We chose ear-level as the unit of analysis because:

1. LFHP is an ear-specific phenomenon, influenced by surgical approach and array choice in that ear

2. The registry data are structured at the ear level, with no prespecified analytic framework

for handling bilateral cases. By design we only know whether a patient is bilaterally

implanted if CIs are included for both ears (thus, if the other ear had a non-MED-EL device

or was implanted in a clinic that does not participate in the registry, we would not know

that the person is a bilateral CI user),

3. From what we do know, the proportion of bilateral recipients is relatively modest (6.1%

VC; 11.6% AAO)

We consider therefore our main findings as not meaningfully altered by bilateral status or

analytic choice, while remaining aligned with the descriptive scope of our study.”

My assessment:

I agree with the authors, but I have a suggestion for further analysis, though probably only in the future when more bilateral datasets are available. The surgeon only manipulates the peripheral auditory system. Effects resulting from bilateral connections are not to be expected. The authors rightly argue that the necessary data is not available with the required clarity. This is also because it would be necessary to consider not only a cochlear implant (CI) on the contralateral side, but also residual hearing and/or hearing aid use. This data would be essential for any analysis. Thank you. We agree and would be happy to consider this in future analyses.

Nevertheless, I find another effect very interesting. To investigate this, there are probably not yet enough datasets. If a CI is implanted on the second side, is the preservation of residual hearing comparable to that on the first side, or not? In other words, if the residual hearing on the first side is well preserved, does the patient then have a better chance on the second side? And if residual hearing is lost on the first side and no unusual occurrence is noted intraoperatively (which is naturally not recorded in the registry), does the patient then also have an increased risk on the second side? Because it may also be that factors such as predisposition or the nature of the underlying disease play a role, which have so far been recorded very little. This is a very interesting topic. We are considering investigating this, but to do so, a more controlled investigation would make more sense.

c)

Reviewer 2:

“Later windows have fewer cases and potential return-bias, complicating interpretation of crosssectional proportions at 6–12/12–24/24–36 months. Fit a repeated-measures mixed-effects model for threshold change (VC bands) using all available visits. For AAO (<80 dB HL), add a timeto- event analysis (e.g., Kaplan–Meier to “loss of functional LFHP”) based on existing timestamps/interval bins, incorporating center fixed effects or a patient-level frailty term where feasible. These steps clarify temporal dynamics under missingness without requiring new data collection.

Author’s response:

“This is a registry study and, as such, looks at real world data to provide real world evidence. As a limiting consequence, participant numbers vary across follow-up intervals.

This reflects typical clinical practice: some patients do not come back, especially if they

experience no issues. Other patients opt of follow ups at trained hearing aid centres instead of the participating clinics, as this can be more convenient in Germany.”

My assessment:

I agree with the authors. In my opinion, the purpose of the study did not require these statistics; rather, these statistics would give a false impression of accuracy.

Thank you for your support on this.

d)

Reviewer 2:

“FLEXSOFT shows moderate “preservation” by VC but low “functional preservation” by AAO—a clinically meaningful divergence rooted in different frequency ranges/thresholds. Where both metrics are available at the same visit, present a cross-tabulation (VC category × AAO

---

## [Decision Letter · Decision Letter 2]

4 Mar 2026

Functional hearing and low frequency hearing preservation after cochlear implant surgery is achievable with FLEX electrode arrays: Real world evidence from the MEHS Registry

PONE-D-25-39581R2

Dear Dr. Polterauer-Neuling,

We’re pleased to inform you that your manuscript has been judged scientifically suitable for publication and will be formally accepted for publication once it meets all outstanding technical requirements.

Kind regards,

Toru Miwa

Academic Editor

PLOS One

Additional Editor Comments (optional):

Reviewers' comments:

Reviewer's Responses to Questions

**Comments to the Author**

Reviewer #2: All comments have been addressed

2. Is the manuscript technically sound, and do the data support the conclusions?

Reviewer #2: Yes

3. Has the statistical analysis been performed appropriately and rigorously?

Reviewer #2: N/A

4. Have the authors made all data underlying the findings in their manuscript fully available?

Reviewer #2: Yes

5. Is the manuscript presented in an intelligible fashion and written in standard English?

Reviewer #2: Yes

Reviewer #2: Thank you for the thorough revision. My main concerns from the previous round have been addressed satisfactorily. In particular, the manuscript now more clearly frames the work as a descriptive registry-based analysis with appropriate limitations, and the additional comparison with prior reports helps contextualize the findings and clarify the contribution of this study.

I also appreciate the revisions prompted by the Editor’s assessment, especially the clearer handling of how VC and AAO are used and how concordance/discordance is reported in a descriptive manner.

Overall, I believe the manuscript is suitable for publication.

.

Reviewer #2: No

---

## [Editor Report · Acceptance letter]

PONE-D-25-39581R2

PLOS One

Dear Dr. Polterauer-Neuling,

I'm pleased to inform you that your manuscript has been deemed suitable for publication in PLOS One. Congratulations! Your manuscript is now being handed over to our production team.

Kind regards,

on behalf of

Dr. Toru Miwa

Academic Editor

PLOS One